# Fetal death and its association with indicators of social inequality: 20-year analysis in Tacna, Peru

Cesar Copaja-Corzo[1]*, Sujey Gomez-Colque[2], Jennifer Vilchez-Cornejo[3,4], Miguel Hueda-Zavaleta[5,6], Alvaro Taype-Rondan[7,8]

**1** Universidad Científica del Sur, Lima, Perú, **2** Universidad Nacional Jorge Basadre Grohmann, Tacna, Perú, **3** Facultad de Salud Pública y Administración, Unidad de Investigación de Enfermedades Emergentes y Cambio Climático, Universidad Peruana Cayetano Heredia, Lima, Perú, **4** Facultad de Medicina Humana, Universidad Nacional de Ucayali, Ucayali, Perú, **5** Facultad de Ciencias de la Salud, Universidad Privada de Tacna, Tacna, Perú, **6** Hospital Daniel Alcides Carrión, EsSalud, Tacna, Perú, **7** Unidad de Investigación para la Generación y Síntesis de Evidencias en Salud, Vicerrectorado de Investigación, Universidad San Ignacio de Loyola, Lima, Perú, **8** EviSalud—Evidencias en Salud, Lima, Perú

* Ccopaja@cientifica.edu.pe

**Data Availability Statement:** The data cannot be shared publicly because they were obtained by request to the HHUT Institutional Research Ethics Committee, who approved access to the data to the

## Abstract

### Objective

The aim of this study was to evaluate the rates of fetal mortality in a Peruvian hospital between 2001 and 2020 and to investigate the association of indicators of social inequality (such as access to prenatal care and education) with fetal mortality.

### Methodology

We conducted a retrospective cohort study, including all pregnant women who attended a Peruvian hospital between 2001 and 2020. We collected data from the hospital's perinatal computer system. We used Poisson regression models with robust variance to assess the associations of interest, estimating adjusted relative risks (aRR) and their 95% confidence intervals (95% CI).

### Results

We analyzed data from 67,908 pregnant women (median age: 26, range: 21 to 31 years). Of these, 58.3% had one or more comorbidities; the most frequent comorbidities were anemia (33.3%) and urinary tract infection (26.3%). The fetal mortality ratio during the study period was 0.96%, with the highest rate in 2003 (13.7 per 1,000 births) and the lowest in 2016 (6.1 per 1,000 births), without showing a marked trend. Having less than six (aRR: 4.87; 95% CI: 3.99–5.93) or no (aRR: 7.79; 6.31–9.61) prenatal care was associated with higher fetal mortality compared to having six or more check-ups. On the other hand, higher levels of education, such as secondary education (aRR: 0.73; 0.59–0.91), technical college (aRR: 0.63; 0.46–0.85), or university education (aRR: 0.38; 0.25–0.57) were associated with a lower risk of fetal death compared to having primary education or no education. In addition, a more recent year of delivery was associated with lower fetal mortality.

authors develop this specific research work. Data is available upon request from the HHUT Institutional Ethics Committee (contact via (052) 583730, email uadi@hospitaltacna.gob.pe and address: Blondell Street S/n, Tacna, Peru) for researchers who meet the criteria for access to sensitive data.

**Funding:** The Universidad Cientifica del Sur finances the article processing charge. The funders had no role in study design, data collection and analysis, decision to publish, or preparation of the manuscript.

**Competing interests:** The authors have declared that no competing interests exist.

## Conclusion

Our study presents findings of fetal mortality rates that are comparable to those observed in Peru in 2015, but higher than the estimated rates for other Latin American countries. A more recent year of delivery was associated with lower fetal mortality, probably due to reduced illiteracy and increased access to health care between 2000 and 2015. The findings suggest a significant association between indicators of social inequality (such as access to prenatal care and education) with fetal mortality. These results emphasize the critical need to address the social and structural determinants of health, as well as to mitigate health inequities, to effectively reduce fetal mortality.

## Introduction

Peru is a country with significant social inequality, with distinct groups that vary in terms of geography and access to economic resources [1]. This inequality leads to individuals living in unfavorable conditions and facing barriers to accessing basic health services [2,3]. Social inequity and barriers to access to basic services can give rise to various health problems. Among these, fetal mortality stands out as an important public health problem.

According to the World Health Organization (WHO), fetal death is defined as the death of a fetus weighing 500 grams or more, or with a gestational age of 20 weeks or more [4,5]. The global prevalence of fetal death was around 2% in 2015 [6]. In 2021, Peru reported a fetal mortality rate of 9.4 per 1,000 live births [7], which is higher than the rate reported for Latin America, which is 8.2 per 1,000 live births [6]. Despite a global decline in the fetal mortality rate from 1990 to 2010, Peru has not seen the same reduction in the fetal mortality rate as other Latin American countries [8,9]. This is particularly concerning as reducing fetal mortality is an explicit objective of the Millennium Development Goals [8,9] and fetal mortality is a crucial indicator of the quality of maternal and child healthcare [9,10].

In Peru, just two studies have evaluated recent trends in fetal death. One study of 177 pregnant women in two hospitals in the Lambayeque region found that working outside the home, a history of previous abortions, and lower levels of education were associated with a higher risk of fetal death, but it was unable to determine a clear trend in fetal mortality over the years [11]. Another study conducted at a Peruvian hospital, which evaluated 17,869 pregnant women in 2003, reported a fetal mortality rate of 7.33 per 1,000 births and found an association between fetal death and maternal health conditions such as anemia and preeclampsia [12].

Social determinants of health, including socioeconomic conditions, could significantly impact fetal death. Understanding how certain indicators of social inequality may contribute to fetal death can help identify populations at higher risk and inform the development of public policies aimed at reducing its consequences. This research aims to evaluate the association between indicators of social inequality (such as access to prenatal care and education) in fetal mortality; we also seek to know the trends in fetal mortality between 2001 and 2020 in a Peruvian hospital.

## Material and methods

### Design and context

We conducted a retrospective cohort study at the Hospital Hipólito Unanue de Tacna (HHUT) (Tacna, Peru) between January 1, 2001 and December 31, 2020. We adhered to the

guidelines set forth by Strengthening the Reporting of Observational Studies in Epidemiology (STROBE) [13] for reporting observational studies.

HHUT is a hospital located in Tacna, a region in southern Peru that has a population of 346,000 inhabitants and is known as a high international trade zone [14]. It is the largest referral hospital in Tacna, and belongs to the Peruvian Ministry of Health (MINSA), serving patients affiliated with the Seguro Integral de Salud.

## Population

We included all pregnant women who were hospitalized in the gynecology and obstetrics service of the HHUT and who, during the study period, had a live or a non-live birth. We excluded pregnant women who were still hospitalized at the time of data collection, pregnant women referred to another facility, and cases where the outcome at the end of gestation was unknown.

We used the study by Kayode G. et al. [15] to calculate statistical power. In this research, the exposure variable was pregnants not educated. Of the pregnant women with adequate education group, 3.4% had fetal death during follow-up, while those without education had 6.4% fetal death. Likewise, a non-exposed/exposed ratio of 1.75 (3284/1866) was reported. With these parameters, a confidence level of 95% and 67,908 participants in the sample of this study, a statistical power of 99.9% was calculated.

## Procedures and variable definition

We collected data using the Perinatal Informatics System of MINSA-Peru, which is a standardized data collection sheet used to gather information on the pregnant woman and the product from the first prenatal care until discharge after delivery [16]. Information was collected by the obstetrician in charge of the evaluation during prenatal care until the end of pregnancy. If the pregnant woman did not attend the check-ups, the information was collected at the end of pregnancy. The epidemiology staff at HHUT compiled the information obtained by the obstetrician and entered it into an Excel spreadsheet. The staff of obstetrics and epidemiology were regularly trained in the quality area of HHUT to ensure proper recording and collection of data.

To collect data for our study, we first obtained permission from the management of HHUT and the Epidemiology area of HHUT. We then trained a member of the research team to collect the information related to our variables of interest from the databases of the Epidemiology area of HHUT.

## Outcome variable

The outcome of interest was fetal death, defined as the death of a fetus that has reached a birth weight greater than or equal to 500 grams, or if birth weight was not available, a gestational age of 20 weeks or crown-to-heel length of 25 cm [5].

## Exposure variables

Social inequality, as defined by WHO, is represented through social stratification, where individuals are placed in different positions based on their income level, poverty, educational attainment, and occupational status [17]. In our study, we focused on two areas: access to health and education [18].

Access to health was evaluated using the number of prenatal care. A pregnant woman was considered to be at high risk of fetal death if she had no prenatal care, at risk if she had one to five prenatal care, and well-controlled if she had six or more prenatal care, according to

MINSA-Peru [19]. Prenatal care, according to MINSA-Peru, is carried out in the following period: first prenatal care (Before 14 weeks), second prenatal care (Before 22 weeks), third prenatal care (Between 22 and 24 weeks), fourth prenatal care (Between week 27 and 29), fifth prenatal care (Between week 33 and 35) and sixth prenatal care (Between week 37 and 41)

Access to education was assessed by the level of education of the pregnant woman and categorized as: university education, higher technical education, secondary education, primary education, or no education [20].

We also collected other maternal characteristics and categorized them based on previous studies or international consensus. These characteristics included: occupation (worker, student and housewife), marital status (single, cohabiting and married), age of the pregnant woman (10 to 19, 20 to 34, and 35 or older) [21], presence of any comorbidities (no/yes), body mass index (BMI) (normal, underweight, overweight and obesity) [22], and whether the woman had any living children (no/yes).

We also collected fetal characteristics and categorized them as: gestational age at clinical evaluation (very preterm, preterm, term, and post-term) [23], weight of the fetus (very low weight, low weight, normal, and macrosomia) [24], and whether there was any malformation (no/yes).

## Statistical analysis

We downloaded the database in a Microsoft Excel document and then exported it for analysis in the statistical program STATA v16. We used frequencies, percentages, central tendency, and dispersion measures to describe the variables.

We evaluated factors associated with fetal death using Poisson regression models with robust variance, estimating the crude (RR) and adjusted (aRR) relative risks and their respective 95% confidence intervals (95% CI). To determine which variables would be included in the fitted model, we used a Directed Acyclic Graph (DAG) (**S1 Fig**). We assessed the variables of educational level and prenatal care and adjusted for maternal age, marital status, maternal comorbidity, maternal BMI, number of living children, history of fetal death, congenital anomalies, and year of delivery.

In addition, we calculate fetal mortality (number of fetal deaths out of total births) for every 1,000 births, and we calculate the percentage of adequate prenatal care and the percentage of pregnant women with higher technical or university education and present it as a linear figure following the data distribution for each year from 2001 to 2020 (**Fig 1A–1C**).

## Ethics

This study was conducted following the international research ethics guidelines of the Declaration of Helsinki. The research protocol was evaluated and approved by the ethics committee of the HHUT and the Epidemiology area of HHUT (identification code: 405-2020-GGR/GOB. REG.TACNA). Informed consent was not requested due to the observational and retrospective nature of the study.

## Results

### Characteristics of the population

We collected information from 68,174 pregnant women during the study period. Of these, 234 were excluded due to delivery outside the institution and 32 were referred to another institution. The final sample size for analysis consisted of 67,908 pregnant women. The median age of the participants was 26 years (IQR 21 to 31), with 66.2% (n = 44 455) not being employed

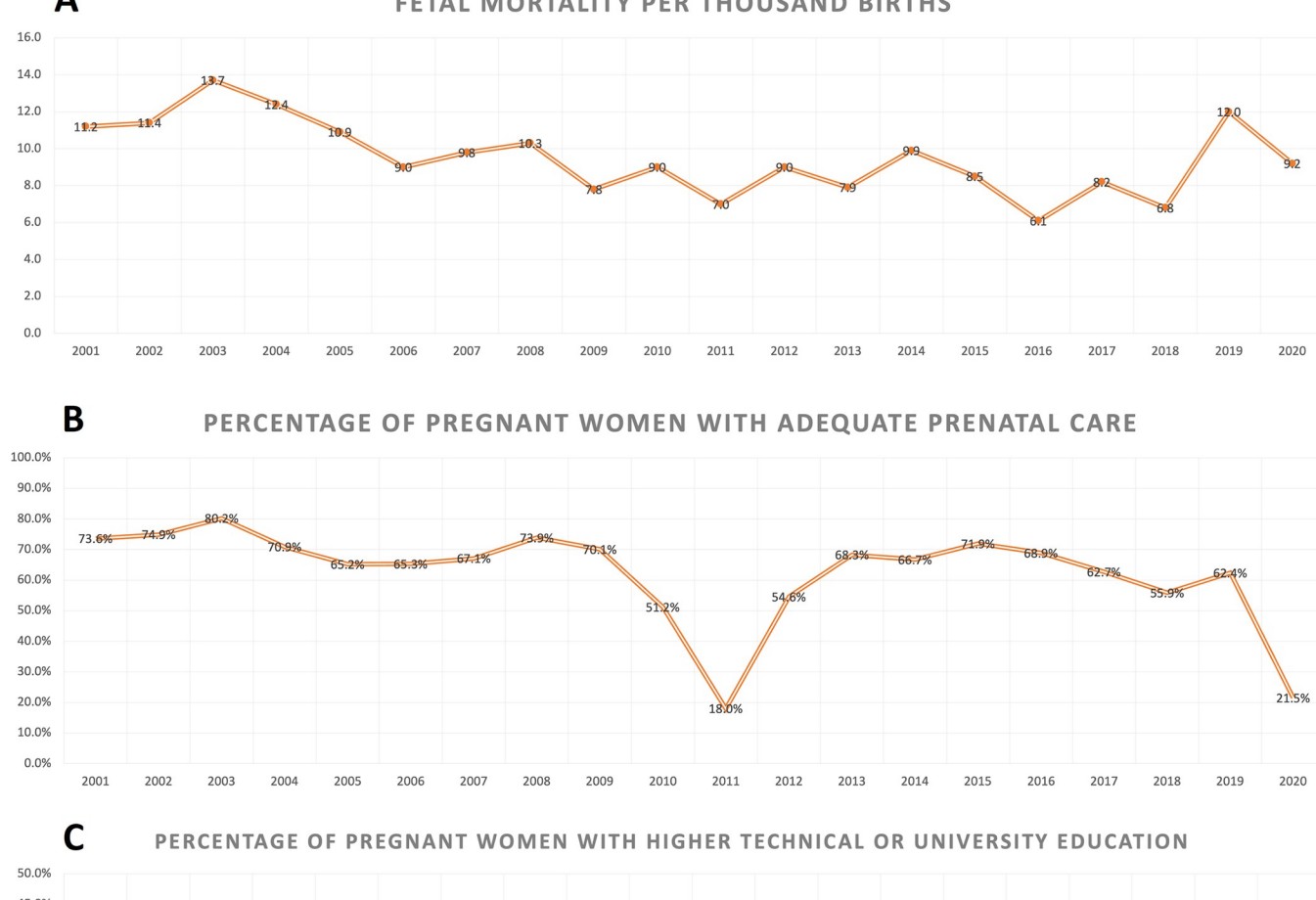

**Fig 1.** A-C: Distribution of fetal mortality and percentage of pregnant women with adequate prenatal care and higher technical or university education, between 2001 and 2020.

and 70.9% (n = 47 957) living with a partner. Additionally, 62.4% (n = 42 391) of the pregnant women had six or more prenatal care.

Regarding the clinical characteristics of the pregnant women, 58.3% (n = 39 611) had at least one comorbidity, with anemia 33.3% (n = 22 583) and urinary tract infection 26.3% (n = 17 835) being the most prevalent. Additionally, 55.3% (n = 37 517) reported having at least one living child, and 1.2% (n = 808) reported having had at least one previous fetal death.

We found a median birth weight of 3500 grams (range 3180 to 3800), with 94% (n = 63 635) being born at term and only 0.3% (n = 224) having congenital anomalies. During the study period, 649 (0.96%) fetuses died (**Table 1**).

**Table 1. Characteristics of the study population (n = 67,908).**

| Characteristics | n (%) |
|---|---|
| Female birth | 34,769 (51.3) |
| Age in years * | 26 (21–31) |
| 10 to 19 | 10,445 (15.4) |
| 20 to 34 | 48,483 (71.4) |
| 35 or more | 8,949 (13.2) |
| Academic degree | |
| No studies or primary | 7,268 (10.7) |
| Secondary studies | 44,989 (66.4) |
| Technician studies | 9,565 (14.1) |
| University studies | 5,899 (8.7) |
| Marital status | |
| Cohabitant | 47,957 (70.9) |
| Married | 9,828 (14.5) |
| Single | 9,848 (14.6) |
| Occupation | |
| Student | 5,845 (8.7) |
| Worker | 16,908 (25.2) |
| Housewife | 44,455 (66.2) |
| Maternal Body Mass Index * | 25.2 (22.9–28.4) |
| Underweight ($< 20$ kg/m$^2$) | 2,957 (4.5) |
| Normal (20 to 24.9 kg/m$^2$) | 28,877 (43.5) |
| Overweight (25 to 29.9 kg/m$^2$) | 23,469 (35.3) |
| Obesity ($\geq 30$ kg/m$^2$) | 11,128 (16.8) |
| Had children who are still alive | 37,517 (55.3) |
| Multiple gestations | 793 (1.17) |
| History of fetal death | 808 (1.2) |
| Prenatal care * | 6 (3–8) |
| Adequate prenatal care ($\geq 6$) | 42,391 (62.4) |
| Poor prenatal care (1 to 5) | 12,943 (19.1) |
| No prenatal care | 12,574 (18.5) |
| Comorbidity during pregnancy | 39,611 (58.3) |
| Anemia | 22,583 (33.3) |
| Urinary infection | 17,835 (26.3) |
| Pregnancy induced hypertension | 1698 (2.5) |
| Syphilis | 221 (0.3) |
| Gestational diabetes | 74 (0.1) |
| Gestational age * | 39 (39–40) |
| Very Preterm (20 to 31 weeks) | 850 (1.3) |
| Preterm (32 to 36 weeks) | 3,064 (4.5) |
| At term (37 to 41 weeks) | 63,635 (94.0) |
| Post-term (more than 42 weeks) | 128 (0.2) |
| Birth weight * | 3500 (3180–3800) |
| Very low weight (500 to 1500 g) | 858 (1.3) |
| Low weight (1501 to 2500 g) | 2,444 (3.6) |
| Normal (2501 to 4000 g) | 55,590 (82.0) |
| Macrosomal ($\geq 4000$ g) | 8,895 (13.1) |
| Birth with congenital anomalies | 224 (0.3) |

(*Continued*)

**Table 1.** (Continued)

| Characteristics | n (%) |
|---|---|
| Year of delivery * | 2010 (2006–2015) |
| 2001 to 2005 | 16,359 (24.1) |
| 2006 to 2010 | 17,962 (26.5) |
| 2011 to 2015 | 17,278 (25.4) |
| 2016 to 2020 | 16,309 (24.0) |
| Fetal death | 649 (0.96) |

* Median (interquartile range).

## Evolution of social inequality over time

The fetal mortality rate per 1,000 births was highest in 2003 at 13.7, and lowest in 2016 at 6.1 (Fig 1A). The years with the lowest proportion of adequate prenatal care were 2011 and 2020, at 18% and 21.5%, respectively (Fig 1B). Additionally, the percentage of pregnant women with higher technical or university education increased over the years, with a peak in 2019 at 29.2%, but a decline in 2020 at 24.5% (Fig 1C).

## Social inequality associated with fetal death

In the adjusted regression analysis, it was found that inadequate prenatal care (aRR: 4.87; 95% CI: 3.99 to 5.93) or no prenatal care (aRR: 7.79; 95% CI: 6.31 to 9.61) were associated with a higher risk of fetal death compared to pregnant women who had adequate prenatal care. Conversely, pregnant women with secondary education (aRR: 0.73; 95% CI: 0.59 to 0.91), technical higher education (aRR: 0.63; 95% CI: 0.46 to 0.85), and university higher education (aRR: 0.38; 95% CI: 0.25 to 0.57) had a lower risk of fetal death compared to those with only primary education or no education. In addition, a more recent year of delivery was associated with lower fetal mortality (**Table 2**).

## Discussion

### Prevalence of fetal death

In our study, the fetal mortality rate was 9.5 per 1,000 births between 2001 and 2020, which is similar to the estimated rate for all of Peru in 2015 (9.0 per 1,000 births) [25]. Fetal mortality rates can vary greatly depending on the socioeconomic and educational level of the country where the study is carried out. For example, a study conducted in nine hospitals in South Africa involving 9,811 pregnant women reported a fetal mortality rate of 15.5 per 1,000 births between 2017 and 2020 [26], while another study in a hospital in Singapore involving 157,039 pregnant women between 2004 and 2016 reported a fetal mortality rate of 5.1 per 1,000 births [27].

The fetal mortality rate reported in our study is above the estimated rate for Latin America in 2015, which was 8.2 per 1,000 births [25], but below the rates reported in low-income countries such as South Africa [26] and Ethiopia [28] which reported rates of 15.5 and 20 per 1,000 births, respectively.

In Peru, various public policies have been implemented to improve maternal and fetal health, such as increasing the frequency of complete prenatal care and institutional deliveries, as well as the progressive improvement of access to essential sanitation services [3]. These

**Table 2. Crude and adjusted regression model to determine risk factors for fetal death (n = 67 908).**

| Characteristic | Live birth | Fetal death | Crude RR (95% CI) | Adjusted RR (95% CI) |
|---|---|---|---|---|
| Prenatal care | | | | |
| Adequate prenatal care | 42,227 (99.6) | 164 (0.4) | Ref | Ref |
| Poor prenatal care | 12,697 (98.1) | 246 (1.9) | **4.91 (4.04 to 5.98)** | **4.87 (3.99 to 5.93)** |
| No prenatal care | 12,335 (98.1) | 239 (1.9) | **4.91 (4.03 to 5.99)** | **7.79 (6.31 to 9.61)** |
| Academic degree of study | | | | |
| No studies or primary | 7,157 (98.5) | 111 (1.5) | Ref | Ref |
| Secondary | 44,558 (99.0) | 431 (1.0) | **0.63 (0.51 to 0.77)** | **0.73 (0.59 to 0.91)** |
| Senior technician | 9,491 (99.2) | 74 (0.8) | **0.08 (0.38 to 0.68)** | **0.63 (0.46 to 0.85)** |
| Academic | 5,869 (99.5) | 30 (0.5) | **0.07 (0.22 to 0.50)** | **0.38 (0.25 to 0.57)** |
| Maternal age | | | | |
| 20 to 34 | 48,074 (99.2) | 409 (0.8) | Ref | Ref |
| 10 to 19 | 10,344 (99.0) | 101 (1.0) | 1.15 (0.92 to 1.42) | 0.91 (0.72 to 1.17) |
| 35 to more | 8,812 (98.5) | 137 (1.5) | **1.81 (1.50 to 2.20)** | **1.60 (1.30 to 1.97)** |
| Marital status | | | | |
| Cohabitant | 47,527 (99.1) | 430 (0.9) | Ref | Ref |
| Married | 9,719 (98.9) | 109 (1.1) | 1.24 (1.00 to 1.52) | **1.26 (1.01 to 1.57)** |
| Single | 9,740 (98.9) | 108 (1.1) | 1.22 (0.99 to 1.51) | 1.11 (0.89 to 1.39) |
| Maternal comorbidity | | | | |
| No | 28,191 (99.6) | 106 (0.4) | Ref | Ref |
| Yes | 39,068 (98.6) | 543 (1.4) | **3.66 (2.97 to 4.50)** | **4.66 (3.79 to 5.73)** |
| Body mass index | | | | |
| Normal | 28,596 (99.0) | 281 (1.0) | Ref | Ref |
| Underweight | 2,929 (99.1) | 28 (1.0) | 0.97 (0.66 to 1.43) | 0.91 (0.62 to 1.34) |
| Overweight | 23,244 (99.0) | 225 (1.0) | 0.99 (0.83 to 1.17) | 0.98 (0.82 to 1.17) |
| Obesity | 11,025 (99.1) | 103 (0.9) | 0.95 (0.76 to 1.19) | 0.93 (0.73 to 1.17) |
| Number of living children | | | | |
| No living children | 30,124 (99.1) | 267 (0.9) | Ref | Ref |
| With living children | 37,135 (99.0) | 382 (1.0) | 0.09 (0.99 to 1.35) | 0.83 (0.69 to 1.02) |
| History of fetal death | | | | |
| No | 66,475 (99.1) | 625 (0.9) | Ref | Ref |
| Yes | 784 (97.0) | 24 (3.0) | **3.19 (2.13 to 4.77)** | **2.99 (1.99 to 4.49)** |
| Congenital anomalies | | | | |
| No | 67,039 (99.1) | 645 (1.0) | Ref | Ref |
| Yes | 220 (98.2) | 4 (1.8) | 1.87 (0.71 to 4.96) | 1.79 (0.68 to 4.70) |
| Year of delivery | | | | |
| 2001 to 2005 | 16,161 (98.8) | 198 (1.2) | Ref | Ref |
| 2006 to 2010 | 17,796 (99.1) | 166 (0.9) | **0.76 (0.62 to 0.94)** | **0.70 (0.57 to 0.86)** |
| 2011 to 2015 | 17,131 (99.2) | 147 (0.9) | **0.70 (0.57 to 0.87)** | **0.65 (0.52 to 0.81)** |
| 2016 to 2020 | 16,171 (99.2) | 138 (0.9) | **0.70 (0.56 to 0.87)** | **0.42 (0.33 to 0.54)** |

The variables of interest were adjusted for maternal age, marital status, maternal comorbidity, maternal BMI, number of living children, history of fetal death, congenital anomalies, and year of delivery.

measures have impacted fetal mortality at the national level, with an estimated 2.8% decrease each year between 2000 and 2015 [25]. However, in our study, we did not observe a clear trend in the reduction of fetal mortality. This may be due to the fact that these public policies were heterogeneously implemented across the country.

## Level of education

In our study, we found that women with higher levels of education had a lower risk of fetal death during pregnancy, which is consistent with the findings of previous research [20,29]. Education level is a social determinant of health that can have a significant impact on mortality levels, as it influences the overall health of women [30], their quality of life, and their economic and social development [3].

During the study period, we observed a slight trend of increasing technical or university education, but a stagnation between 2016 and 2020, similar to what was reported in another study in Peru in 2019 [31]. In Peru, illiteracy is higher in rural areas (14.5%), among women (8.1%), and in the poorest population (quintile I) (14.2%), reaching 20.9% among poor women [32]. This is particularly relevant as education level is inversely related to poverty, which is also a social determinant of health and can influence fetal mortality [33].

The proportion of the Peruvian population living in poverty has significantly decreased from 13.5% to 3.5% between 2006 and 2016, placing Peru below the average for Latin America and the Caribbean but still above other regions of the world [18,34]. This decrease in poverty may have contributed to a decrease in fetal mortality in recent years. The gap in access to education may have been exacerbated by COVID-19, as the lack of access to the Internet and inadequate digital skills have prevented many people, especially the most vulnerable, from working or studying from home [33]. This issue must continue to be addressed to promote equal opportunities, particularly for Peruvian women [3].

## Prenatal care

Prenatal care is crucial for the health of both the mother and child. WHO reports that inadequate prenatal care increases the risk of complications during pregnancy [35]. This could explain why the risk of fetal death in our study was substantially increased in pregnant women with no prenatal care or fewer than six care. Despite free care, some pregnant women, particularly those who are uneducated, poor, and from ethnic minority groups, often seek prenatal care late in their pregnancy [20].

In addition, our study found that the percentage of adequate prenatal care remained above 50% in most years, but there was a marked decrease in adequate prenatal care in 2020. This could be attributed to the state of emergency and mandatory social isolation declared in Peru due to the COVID-19 pandemic, which greatly affected Peru's first level of care. Mandatory isolation could have caused pregnant women not to attend their prenatal care due to the restrictions imposed and for fear of contagion from COVID-19. In addition, due to the state of emergency decreed in Peru, all health services focused on containing the pandemic; this could also have decreased the control and monitoring of pregnant women during the pandemic.

## Year of delivery

In the adjusted analysis, we found that a more recent year of delivery was associated with lower fetal mortality. This suggests a cohort effect, which may be due to the reduction in the percentage of illiteracy, which decreased from 8.3% to 0.3% in Tacna, Peru, between 2000 and 2015 [36,37]. On the other hand, the percentage of pregnant women with three controls prenatal or less decreased from 7% to 4.2% between the years 2000 and 2015, showing improved access to health care [36,37]. These findings align with global efforts to improve reproductive health care and highlight the potential impact of better interventions to improve maternal and fetal health outcomes. However, it is important to recognize potential limitations, such as unmeasured factors or confounding variables, that could have influenced the observed association.

## Implications

The Peruvian health system have to identify barriers to access to health services and improve the coverage or capacity of the system to respond to the health needs of the population [34]. Public policies such as Universal Health Insurance have been partially implemented in Peru to reduce this gap, which has led to an increase in the coverage of affiliation to some type of health insurance, with Seguro Integral de Salud increasing from 19.4% to 50.2% between 2006 and 2015 [38]. Barriers to access to health services include education level, residence in a rural area or with geographical inaccessibility, and poverty [39]. In this context, addressing the social and structural determinants of health can reduce social disadvantage and improve health outcomes [40]. This approach can be a highly effective way of addressing health inequities [41].

The present study has certain limitations that should be considered when interpreting its results. The main limitation was its retrospective nature, which prevents the evaluation of certain variables that could be confounding factors or contribute to explaining the study phenomenon (such as socioeconomic status, ethnicity, or quality of life). Furthermore, the information collected from pregnant women during prenatal visits was more exhaustive, unlike pregnant women who did not attend prenatal check-ups. This last information, collected before labor, can be unreliable. Additionally, the study was only conducted in one institution, hence the results are not generalizable to entire Peruvian population. However, this study has important strengths; first of all, to our knowledge, it is the first study that addresses the trend of fetal mortality and its associated factors in the Peruvian population. Furthermore, the large sample size provided the sufficient statistical power (99.9%) to our study.

## Conclusion

In conclusion, our assessment of a hospital located in Tacna, Peru, unveiled that fetal mortality rates were comparable to the 2015 Peruvian average but higher than the estimated rates for other Latin American countries. A more recent year of delivery was associated with lower fetal mortality, probably due to reduced illiteracy and increased access to health care between 2000 and 2015. It is worth noting that inadequate prenatal care and lower levels of education among pregnant women were strongly associated with fetal mortality. These results highlight the importance of addressing health's social and structural determinants and reducing health inequities to reduce fetal mortality.

## Supporting information

**S1 Fig. Directed acyclic diagram to determine the possible causes of fetal death.**
(PDF)

## Author Contributions

**Conceptualization:** Cesar Copaja-Corzo, Sujey Gomez-Colque, Miguel Hueda-Zavaleta.

**Data curation:** Sujey Gomez-Colque.

**Formal analysis:** Cesar Copaja-Corzo, Alvaro Taype-Rondan.

**Funding acquisition:** Cesar Copaja-Corzo.

**Methodology:** Cesar Copaja-Corzo, Jennifer Vilchez-Cornejo, Miguel Hueda-Zavaleta.

**Writing – original draft:** Cesar Copaja-Corzo, Sujey Gomez-Colque, Jennifer Vilchez-Cornejo, Miguel Hueda-Zavaleta.

**Writing – review & editing:** Miguel Hueda-Zavaleta, Alvaro Taype-Rondan.

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
