## [Decision Letter · Decision Letter 0]

30 May 2023

PONE-D-23-03326Fetal death and its association with indicators of social inequality: 20-year cohort analysis in Tacna, PeruPLOS ONE

Dear Dr. Corzo

Thank you for submitting your manuscript to PLOS ONE. After careful consideration, we feel that it has merit but does not fully meet PLOS ONE’s publication criteria as it currently stands. Therefore, we invite you to submit a revised version of the manuscript that addresses the points raised during the review process.

We look forward to receiving your revised manuscript.

Kind regards,

Zahra Hoodbhoy

Academic Editor

PLOS ONE

Journal Requirements:

Reviewers' comments:

Reviewer's Responses to Questions

**Comments to the Author**

1. Is the manuscript technically sound, and do the data support the conclusions?

Reviewer #1: Yes

2. Has the statistical analysis been performed appropriately and rigorously? 

Reviewer #1: No

3. Have the authors made all data underlying the findings in their manuscript fully available?

Reviewer #1: No

4. Is the manuscript presented in an intelligible fashion and written in standard English?

Reviewer #1: No

5. Review Comments to the Author

Reviewer #1: Overall, the manuscript and analysis are feasible and scientifically make sense but the analysis itself is not clear, especially with the addition of "fetal death" as a confounder when it is an outcome also. English needs to be heavily revised.

6. PLOS authors have the option to publish the peer review history of their article (what does this mean?). If published, this will include your full peer review and any attached files.

Reviewer #1: No

---

## [Author Response · Author response to Decision Letter 0]

19 Jul 2023

Response to reviewers: 

We appreciate your review. Your comments enhance the depth of our manuscript. Comments are answered below:

Abstract

Line 30. Sentence should state: “we conducted a retrospective cohort study” …

- Thank you, we made the suggested modification in the abstract, as follows: “Methodology: We conducted a retrospective cohort study”

Line 35. Median age should accompany a range. 

- We add the age range in line 35, as follows: “(median age: 26, range: 21 to 31 years).”

Line 35. Sentence should not begin with numerals. Either reword the sentence or spell out the numerical value.

- We follow your suggestion and modify the wording, as follows: “Of these, 58.3% had one or more comorbidities”

Line 35-37. Please revise – grammatically incorrect.

- We made changes to improve the wording, as follows: “We analyzed data from 67,908 pregnant women (median age: 26, range: 21 to 31 years). Of these, 58.3% had one or more comorbidities; the most frequent comorbidities were anemia (33.3%) and urinary tract infection (26.3%).”

Line 37-38. Add fetal mortality rate for 2003 and for 2016. 

- We add the information, as follows: “2003 (13.7 per 1,000 births) and the lowest in 2016 (6.1 per 1,000 births)”

Line 39. All numbers less than or equal to 10 should be spelled out.

- We have corrected throughout the manuscript.

Line 44-48. Please revise the conclusion - grammatical errors.

- You're right, we made the pertinent change in that section, as follows: “Our study presents findings of fetal mortality rates that are comparable to those observed in Peru but higher than the estimated rates for other Latin American countries. A more recent year of delivery was associated with lower fetal mortality. The findings suggest a significant association between inadequate prenatal care, lower levels of education, and elevated rates of fetal mortality. These results emphasize the critical need to address the social and structural determinants of health, as well as to mitigate health inequities, to effectively reduce fetal mortality.”

Introduction

Lines 53-57: Introduction overall is meaningful but paragraphs 1 and 2 are not connected. Please revise and add in a transition. 

- You are correct; we have added a transition between both themes, as follows: “Social inequity and barriers to access to basic services can give rise to various health problems. Among these, fetal mortality stands out as an important public health problem.”

Line 56: Sounds like you’re using the definition of stillbirth by the WHO – can you please clarify this. It’s important to clarify how fetal death is defined b/c based on the cutoffs and how it’s defined, it can be considered stillbirth or miscarriage. Table 1 in your reference (Fernanda Tavares Da Silva, 2016) lists the definition you’re using by the WHO as stillbirth. In your table 2 below you make a distinction between “no stillbirth” and “fetal death” – which can be two different things depending on the definition. 

- You are correct; we have modified and clarified using the WHO definition, as follows: “According to the World Health Organization (WHO), fetal death is defined as the death of a fetus weighing 500 grams or more, or with a gestational age of 20 weeks or more”

Lines 57: “The overall prevalence of fetal death is around 2%”. What year is it 2% in? 

- Added the year (2015).

Line 60: “…Peru has not seen the same reduction as other countries…”. Are you referring to the rate of reduction here? Please clarify.

- Yes, we are referring to the fetal mortality rate. Clarification was made, as follows “Despite a global decline in the fetal mortality rate from 1990 to 2010, Peru has not seen the same reduction in the fetal mortality rate as other Latin American countries.”

Line 61: “This is particularly concerning as reducing fetal mortality was an explicit objective of the MDG”. Should state, it is an explicit objective of the MDG. 

- You are right, we made the change.

Line 64: You mentioned “few studies” that have evaluated recent trends in fetal death. Then you mentioned two studies. “few” means three or more studies. Either add another study or you can say something like “just two studies”. 

- You're right; we made the change and mentioned that it was only two studies.

Lines 74-76: It sounds like you are looking at the association between prenatal care and education and fetal mortality. Your sentence makes it sound like you are assessing the impact of fetal mortality on prenatal care and education. Please revise the order. 

- You are correct; we have modified it to clarify the objective of the investigation, as follows: “This research aims to evaluate the association between indicators of social inequality (such as access to prenatal care and education) in fetal mortality; we also seek to know the trends in fetal mortality between 2001 and 2020 in a Peruvian hospital.”

Materials and methods

Line 89: Please refer to a “live or non-live new born” outcome as “birth” instead of “newborn”. Make sure to revise the manuscript for this discrepancy throughout. 

- Five discrepancies were found: 1 in the text, 3 in tables, and 1 in the bibliography. The first four were replaced as suggested.

Lines 94-99: It is unclear exactly what’s collected on the standardized data collection sheet. Is all the information collected by the obstetrician? If information is not collected at the check-up visits, how comprehensive is it at the end of pregnancy? 

- You are right; we have better detailed the content of the data collection sheet. All information about the mother is collected by the obstetrician, and information about the delivery and the product is collected by the attending physician. All the data is collected during the prenatal care; only if the pregnant woman has not attended any prenatal care and finally goes into labor, the obstetrician and the doctor collect all the information that can be obtained at that moment. The records filled out when the pregnant woman doesn't have any prenatal care may have unreliable information; this is a limitation that we added to our study.

Line 117: “A pregnant women was considered to be at high risk…” At risk of what? 

- High risk of fetal death. Added clarification, as follows: “A pregnant woman was considered to be at high risk of fetal death if she had no prenatal care, at risk if she had one to five prenatal care, and well-controlled if she had six or more prenatal care, according to MINSA-Peru.”

Line 127: “thinness” should be listed as “underweight”

- We replaced “thinness” with “underweight” and revised throughout the manuscript.

Q1. Were women treated for comorbidities such as UTI or syphilis? What was the rate of pregnancy induced hypertension or chronic hypertension in this cohort. Adding some of these additional variables to Table 1 would be help clarify.

- All women who developed UTIs or syphilis received treatment. In our cohort, 2.5% had pregnancy-induced hypertension. We added this variable in Table 1.

Q2. Did your data have any multiple gestations? If so, how was the data handled? 

- Yes, in our base, we had the report of multiple gestations. We also included women with multiple pregnancies who experienced fetal death. We added the description of the total number of women with multiple gestations in Table 1.

Q3. At what time points were the prenatal visits done? Please list gestational weeks and how many women attended vs. how much missing data there was.

- Prenatal care was carried out according to MINSA-Peru guidelines, as follows: “first prenatal care (Before 14 weeks), second prenatal care (Before 22 weeks), third prenatal care (Between 22 and 24 weeks), fourth prenatal care (Between weeks 27 and 29), fifth prenatal care (Between weeks 33 and 35) and sixth prenatal care (Between weeks 37 and 41).” 

- In Table 1, we describe the median and interquartile range of the number of prenatal check-ups pregnant women had. On the other hand, we did not have the exact date on which the control was carried out (the exact week of gestation) because the MINSA-Peru guidelines only require ranges for the evaluation of the pregnant woman.

Statistical analysis

Line 140: Please show the DAG that was used as a figure here or as a supplementary figure. 

- We have added the DAG as supplementary material.

Line 141: Why are you adjusting for fetal death when assessing fetal death as an outcome? What is the significant of the “year of delivery” as a confounder? Are you implying some sort of a cohort effect – if so, please clearly discuss that in the Conclusion section. 

- There was an error in the wording; we wanted to mention a history of fetal death, which has already been modified. The year of delivery was taken into account because the Peruvian health system changed over the years, which could influence prenatal care.

- Regarding the year of delivery variable, we have discussed the result in the “year of delivery” subheading of the discussion, as following: “We found that, in the adjusted results, a more recent year of delivery was associated with lower fetal mortality. This suggests a cohort effect, which may be due to advancements in healthcare practices, medical technology, and prenatal care over time. These findings align with global efforts to enhance reproductive healthcare and highlight the potential impact of improved interventions on improving maternal and fetal health outcomes. However, it is important to acknowledge potential limitations, such as unmeasured factors or confounding variables, that could have influenced the observed association.”

Line 144: What graphs are you referring to? Do you mean “figures” Please revise the sentence. You do not need to say “Finally, to develop the graphs…” You can just begin the sentence by saying “we also calculated fetal mortality…” When you discuss this in the Results, you can refer to the figure numbers. 

- We refer to Figure 1 A-C. No, for the distribution of education level and prenatal care, we put percentages. You are right; in the results section, we refer to the numbers on the figures when talking about them.

Results

Line 153: Report median age with range. Also please report n= with percentage of each of the variables mentioned (such as: 66.2% employed (n=?)). 

- Ok, the comments made on the results have been modified following your guidelines.

Line 162: Table one should be stratified by primary exposure. When stratified by exposure, it allows assessment of potential confounding. There might be uneven distribution of these characteristics between exposed and unexposed.

- As we stated in the objective, we aimed to “indicators of social inequality (such as access to prenatal care and education)”. Thus, we did not have a primary exposure

Line 169: Quality of figures is very poor. Hard to read numbers. Please fix Figure 1B title (what is “prenatal controls”? did you mean to say prenatal care?). Please also refer to A B and C in the text in results when presenting numbers. 

- We have corrected and improved the quality of the figure and corrected the term prenatal care in the figure and throughout the entire manuscript.

Line 187: You adjusted your model assessing the risk of fetal death with fetal death? Please clarify which variables were considered confounders. If you ran a t-test on a set of demographic variables, please show the results in Table 1. 

- There was a writing error; we wanted to mention a history of fetal death; we have made the correction. The variables that have been considered as confounding variables are maternal age, marital status, maternal comorbidity, maternal BMI, number of living children, history of fetal death, congenital anomalies, and year of delivery. We did not use t-tests for demographic variables as detailed in the statistical analysis described.

Line 166: In your methods you mentioned prenatal checkups were at their lowest in 2011 and 2020, can you comment on why this might be under the conclusion section.

- Thank you, we decided to talk more about the decline in the percentage of prenatal care in the discussion section, as follow: “Mandatory isolation could have caused pregnant women not to attend their prenatal care due to the restrictions imposed and for fear of contagion from covid-19. In addition, due to the state of emergency decreed in Peru, all health services focused on containing the pandemic; this could also have decreased the control and monitoring of pregnant women during the pandemic.”

Lines 165-169: Would be meaningful to see absolute and relative change between 2011 and 2020. Please create a new table showing this change. 

- We do present the association between year of delivery and fetal mortality in figure 1 and table 2. 

Lines 183-184: Table 2 shows columns “no stillbirth” and “fetal death”. Can you clarify what “no stillbirth” means? 

- We decided to use the term live birth for better understanding.

Conclusion

Line 192: If fetal mortality rates vary based on context, what contexts are you referring to? Please elaborate? 

- You are right; we have detailed the content of the statement better. According to the World Health Organization, the fetal mortality rate is higher in low- and middle-income countries compared to high-income countries; therefore, the fetal mortality rate can vary greatly depending on the country being studied due to the difference in income level, as well as access to education and prenatal care that may exist between different countries.

Line 236: What are you referring to when you say “marked decreased in good prenatal check-ups in 2020”? In this paper, you are assessing the number of prenatal visits? What does “good” refer to her? 

- You are correct; we have made the modification, and we clarify that reference is made to the decrease in adequate prenatal care that occurred in 2020. Considering adequate prenatal care, the performance of 6 or more prenatal care during pregnancy, as indicated in the methodology section and Figure 1.

Lines 247-249: These lines should be merged with the paragraph before? It does not need its own paragraph. 

- You are right; we have followed your suggestion and made the relevant modification in that section.

Lines 250-257: Limitations section is very weak. There is also collection of data retrospectively – mentioned in the methods section. Can you please comment on the impact of data collected at the end of pregnancy vs. at each antenatal visit. 

There is nothing on the strengths of the study, please add.

- You are right; we have clarified that retrospective collection data is a limitation of the study. We also add that the information collected throughout pregnancy during prenatal care was much more complete than that collected at the time of delivery in those pregnant women who did not have prenatal care; the latter information is unreliable. Finally, we add the strengths of our research, as follow “Furthermore, the information collected from pregnant women during prenatal visits was more exhaustive, unlike pregnant women who did not attend prenatal check-ups. This last information, collected before labor, can be unreliable. Additionally, the study was only conducted in one institution, which makes the results unrepresentative of the entire Peruvian population. However, this study has important strengths; first of all, to our knowledge, it is the first study that addresses the trend of fetal mortality and its associated factors in the Peruvian population. It also has a large sample size and a large amount of information about the variables of interest, which helps to have a better vision of reality.”

Lines 259-261: It is very unclear whether fetal mortality trends you observed in your study are in line with the national trends or not. In lines 203-206 in the Results section, you mention: These measures have impacted fetal mortality at the national level, with an estimated 2% decrease each year between 2000 and 2015 [24]. However, in our study, we did not observe a clear trend in the reduction of fetal mortality.” But in the conclusion here, you mention: “In conclusion, in the hospital evaluated, fetal mortality was similar to the Peruvian rate…”. 

Please also revise this sentence per grammatical errors. 

- You are right; we have specified that the Peruvian fetal death trend has decreased, but our study cannot observe this downward trend. We also specify that fetal mortality was like the Peruvian rate but that of 2015. We have made the correction and improvements in the wording, as follow: “In conclusion, our evaluation of the hospital revealed that fetal mortality rates were comparable to the 2015 Peruvian average but higher than the estimated rates for other Latin American countries. A more recent year of delivery was associated with lower fetal mortality. It is worth noting that inadequate prenatal care and lower levels of education among pregnant women were strongly associated with fetal mortality.”

Overall: Writing needs to be revised. Sentences and ideas are not clear. Analysis needs revision especially assessing confounders. 

- We appreciate your valuable contributions, we have carried out an extensive revision in the drafting of the manuscript, and we have corrected the comments raised; we consider that the quality of our manuscript has greatly improved.

---

## [Decision Letter · Decision Letter 1]

21 Aug 2023

PONE-D-23-03326R1Fetal death and its association with indicators of social inequality: 20-year analysis in Tacna, PeruPLOS ONE Dear Dr. Corzo,

Thank you for submitting your manuscript to PLOS ONE. After careful consideration, we feel that it has merit but does not fully meet PLOS ONE’s publication criteria as it currently stands. Therefore, we invite you to submit a revised version of the manuscript that addresses the points raised during the review process.

We look forward to receiving your revised manuscript.

Kind regards,

Zahra Hoodbhoy

Academic Editor

PLOS ONE

Reviewers' comments:

Reviewer's Responses to Questions

**Comments to the Author**

1. If the authors have adequately addressed your comments raised in a previous round of review and you feel that this manuscript is now acceptable for publication, you may indicate that here to bypass the “Comments to the Author” section, enter your conflict of interest statement in the “Confidential to Editor” section, and submit your "Accept" recommendation.

Reviewer #2: All comments have been addressed

2. Is the manuscript technically sound, and do the data support the conclusions?

Reviewer #2: Yes

3. Has the statistical analysis been performed appropriately and rigorously? 

Reviewer #2: Yes

4. Have the authors made all data underlying the findings in their manuscript fully available?

Reviewer #2: No

5. Is the manuscript presented in an intelligible fashion and written in standard English?

Reviewer #2: Yes

6. Review Comments to the Author

Reviewer #2: I reviewed the study titled “Fetal death and its association with indicators of social inequality: 20-year analysis in Tacna, Peru”.

The manuscript has addressed the previous comments satisfactorily. There are few comments that should be addressed before the paper goes for publication:

1. Lines 266-267. Authors mentioned that reduced rate of fetal death in recent years “which may be due to advancements in healthcare practices, medical technology, and prenatal care over time.”

It will be better if authors can comment on the change in literacy level or ANC care seeking in the same years and see if it is aligned with the reduction in fetal mortality. It is suggested because if a significant change can be brought by improving the system delivery alone then how much incremental change can be caused by addressing the social determinants.

2. Line 290- In limitations, “Additionally, the study was only conducted in one institution, which makes the results unrepresentative of the entire Peruvian population.”

It can be rephrased as “Additionally, the study was only conducted in one institution, hence the results are not generalizable to entire Peruvian population.”

3. Lines 292-293 “It also has a large sample size and a large amount of information about the variables of interest, which helps to have a better vision of reality.”

It needs rephrasing. It sounds like a non-scientific statement. Please rephrase and talk about study power and accurate results instead of reality as reality is a subjective term.

4. Line 298. “A more recent year of delivery was associated with lower fetal mortality.”

Add some description why it happened. Just the statement in the conclusion is not informative for the readers.

7. PLOS authors have the option to publish the peer review history of their article (what does this mean?). If published, this will include your full peer review and any attached files.

Reviewer #2: **Yes: **Sana Sheikh

---

## [Author Response · Author response to Decision Letter 1]

28 Aug 2023

We appreciate your review. Your comments enhance the depth of our manuscript. Comments are answered below:

1. Lines 266-267. Authors mentioned that reduced rate of fetal death in recent years “which may be due to advancements in healthcare practices, medical technology, and prenatal care over time.”

It will be better if authors can comment on the change in literacy level or ANC care seeking in the same years and see if it is aligned with the reduction in fetal mortality. It is suggested because if a significant change can be brought by improving the system delivery alone then how much incremental change can be caused by addressing the social determinants.

- Thank you, we follow your suggestion and modify the wording, as follows: “In the adjusted analysis, we found that a more recent year of delivery was associated with lower fetal mortality. This suggests a cohort effect, which may be due to the reduction in the percentage of illiteracy, which decreased from 8.3% to 0.3% in Tacna, Peru, between 2000 and 2015 [35,36]. On the other hand, the percentage of pregnant women with three controls prenatal or less decreased from 7% to 4.2% between the years 2000 and 2015, showing improved access to health care [35,36].”

2. Line 290- In limitations, “Additionally, the study was only conducted in one institution, which makes the results unrepresentative of the entire Peruvian population.” It can be rephrased as “Additionally, the study was only conducted in one institution, hence the results are not generalizable to entire Peruvian population.”

- Thank you, we made the suggested modification as follows: “Additionally, the study was only conducted in one institution, hence the results are not generalizable to entire Peruvian population.”

3. Lines 292-293 “It also has a large sample size and a large amount of information about the variables of interest, which helps to have a better vision of reality.”

It needs rephrasing. It sounds like a non-scientific statement. Please rephrase and talk about study power and accurate results instead of reality as reality is a subjective term.

- Thank you, we follow your suggestion and modify the wording, as follows: “Furthermore, the large sample size and the large amount of information on the variables of interest provide greater statistical power to our study.”

4. Line 298. “A more recent year of delivery was associated with lower fetal mortality.”

Add some description why it happened. Just the statement in the conclusion is not informative for the readers.

- Thank you, we follow your suggestion and modify the wording, as follows: “A more recent year of delivery was associated with lower fetal mortality, probably due to reduced illiteracy and increased access to health care between 2000 and 2015.”

---

## [Decision Letter · Decision Letter 2]

6 Sep 2023

PONE-D-23-03326R2Fetal death and its association with indicators of social inequality: 20-year analysis in Tacna, PeruPLOS ONE

Dear Dr. Corzo,

Thank you for submitting your manuscript to PLOS ONE. After careful consideration, we feel that it has merit but does not fully meet PLOS ONE’s publication criteria as it currently stands. Therefore, we invite you to submit a revised version of the manuscript that addresses the points raised during the review process

We look forward to receiving your revised manuscript.

Kind regards,

Zahra Hoodbhoy

Academic Editor

PLOS ONE

Journal Requirements:

Reviewers' comments:

Reviewer's Responses to Questions

**Comments to the Author**

1. If the authors have adequately addressed your comments raised in a previous round of review and you feel that this manuscript is now acceptable for publication, you may indicate that here to bypass the “Comments to the Author” section, enter your conflict of interest statement in the “Confidential to Editor” section, and submit your "Accept" recommendation.

Reviewer #2: All comments have been addressed

2. Is the manuscript technically sound, and do the data support the conclusions?

Reviewer #2: Yes

3. Has the statistical analysis been performed appropriately and rigorously? 

Reviewer #2: Yes

4. Have the authors made all data underlying the findings in their manuscript fully available?

Reviewer #2: No

5. Is the manuscript presented in an intelligible fashion and written in standard English?

Reviewer #2: Yes

6. Review Comments to the Author

Reviewer #2: Thanks for submitting the revised manuscript. All the comments have been addressed satisfactorily except one.

Authors revised the statement to “Furthermore, the large sample size and the large amount of information on the

variables of interest provide greater statistical power to our study.”

Unless you do a post-hoc power calculation, it cannot be claimed. It is suggested to calculate the power, if it has not been done at the time of sample size estimation and report it. For e.g. the large sample size provided the sufficient statistical power (80%) to our study. Please note that power is only related to sample size and not number of variables or large amount of information.

7. PLOS authors have the option to publish the peer review history of their article (what does this mean?). If published, this will include your full peer review and any attached files.

Reviewer #2: **Yes: **Sana Sheikh

---

## [Author Response · Author response to Decision Letter 2]

13 Sep 2023

Thank you, we follow your suggestion. and calculated the statistical power of our study. We detail this information in the methodology section as follows: “We used the study by Kayode G. et al. [15] to calculate statistical power. In this research, the exposure variable was pregnants not educated. Of the pregnant women with adequate education group, 3.4% had fetal death during follow-up, while those without education had 6.4% fetal death. Likewise, a non-exposed/exposed ratio of 1.75 (3284/1866) was reported. With these parameters, a confidence level of 95% and 67,908 participants in the sample of this study, a statistical power of 99.9% was calculated.” We also detail this information in the Discussion section, on the strengths of the study, and modify the wording, as follows: “…Furthermore, the large sample size provided the sufficient statistical power (99.9%) to our study.”

Additionally, the information added on this and the previous review was properly cited and added to the bibliography (No. 15, 36 and 37). Consequently, we update the reference list.

We consider that after following your recommendations, our study has improved its quality. 

Thank you.

---

## [Editor Report · Decision Letter 3]

14 Sep 2023

Fetal death and its association with indicators of social inequality: 20-year analysis in Tacna, Peru

PONE-D-23-03326R3

Dear Dr. Cora,

We’re pleased to inform you that your manuscript has been judged scientifically suitable for publication and will be formally accepted for publication once it meets all outstanding technical requirements.

Kind regards,

Zahra Hoodbhoy

Academic Editor

PLOS ONE

---

## [Editor Report · Acceptance letter]

28 Sep 2023

PONE-D-23-03326R3 

Fetal death and its association with indicators of social inequality: 20-year analysis in Tacna, Peru 

Dear Dr. Copaja-Corzo:

I'm pleased to inform you that your manuscript has been deemed suitable for publication in PLOS ONE. Congratulations! Your manuscript is now with our production department. 

Kind regards, 

on behalf of

Dr. Zahra Hoodbhoy 

Academic Editor

PLOS ONE